# Changes to Adult Dog Social Behaviour during and after COVID-19 Lockdowns in England: A Qualitative Analysis of Owner Perception

**DOI:** 10.3390/ani12131682

**Published:** 2022-06-29

**Authors:** Holly Boardman, Mark James Farnworth

**Affiliations:** Department of Animal Health, Behaviour and Welfare, Harper Adams University, Newport TF10 8NB, UK; hollyboardman98@gmail.com

**Keywords:** aggression, behaviour, companion animal, COVID-19, separation anxiety

## Abstract

**Simple Summary:**

Throughout the COVID-19 pandemic, dogs have provided their owners with support and comfort, potentially helping owners cope. However, there could also be negative aspects of pet ownership during and after this period as there are fears that problem behaviours, such as aggression, could increase. Potential social behaviour changes throughout the COVID-19 pandemic have been identified but, so far, the focus has been on puppies rather than adult dogs. This study aims to investigate the perceived impact on canine social behaviour from the owner’s perspective. Fifteen owners of dogs aged between 3 and 6 years were recruited for interviews about their dogs’ social behaviours before, during and after lockdown. The results of this study illustrate that many households have faced different experiences. Owners regarded lockdown as a positive experience for their pet dogs; however, many dogs faced difficulties when restrictions were lifted. Emphasis was placed on the negative impact of lockdown with dogs becoming either overexcited or showing fear-related behaviours, particularly following the lockdowns. Further research into the long-term implications of the pandemic is required to understand the social behaviour of dogs more thoroughly.

**Abstract:**

Reports suggest that dogs have experienced more quality time with their owners and have exhibited less separation-related behaviour during COVID-19. This study aims to define and explore the changes in the social behaviour of adult dogs, identify any perceived short- and long-term effects and identify the implications that future events may have due to the implementation and withdrawal of COVID-19 restrictions. Owners of dogs aged between 3 and 6 years living in England were recruited for semi-structured interviews about their dogs’ social behaviours before, during and after lockdown. Interviews were transcribed and thematic analysis was used to identify key themes. Fifteen owners of eighteen dogs were interviewed in December 2021 to January 2022. All owners interviewed noticed a change in behaviour either during or following the COVID-19 lockdowns. Many owners found that fear-related and aggressive behaviours increased, particularly after lockdown restrictions had been lifted. There is a risk to human and animal safety if fear-related behaviours are not properly managed and there is a danger to dog welfare through relinquishment as owners struggle to cope with problematic behaviours. An increase in the duration of walks is comparable to other studies; however, this study found that many owners continued to walk their dogs more than restrictions allowed. Reliable information or behavioural support by qualified professionals may be needed to aid owners in mitigating the impacts of the COVID-19 pandemic. Behaviour modification plans or information could include positive techniques such as desensitisation and counterconditioning. More widely, owners should be prepared for any future behavioural changes due to unpredictable events which may alter the dog–owner dynamic. Results can be adapted to the increase in at-home working following the pandemic or other novel experiences and changes of routine such as retirement, job change or relocation. More awareness of the long-term implications of the COVID-19 pandemic is needed to prepare dog behaviour professionals for future owner concerns.

## 1. Introduction

The COVID-19 pandemic caused by SARS-CoV-2 has impacted people’s lives globally. They have been affected not only by financial and work losses but also by the restriction of social interactions and the requirement to stay at home. The structure of some households changed due to lockdown regulations, with relatives from different households living together to reduce isolation and support the elderly and people with disabilities [1], whilst others lived apart from those family members considered vulnerable or “key workers”. These restrictions had psychological consequences for the well-being of the human population, and dogs may have experienced similar effects. Dogs have potentially helped their owners to cope during this period, acting as a social support and providing comfort [2].

An unexpected consequence of the COVID-19 pandemic was the large increase of new pet owners—with many finding themselves in the position to have a pet for the first time [3]. Consequently, the UK dog population increased to 12 million dogs in 2021 from 9 million in 2019 [4]. Packer et al. [5] found that puppies, in particular, have been in extreme demand, and new owners were less likely to seek credible breeders and to see the puppy in person before collection.

Christley et al. [6] reported that dog owners in the United Kingdom (UK) increased the amount of time training and playing with their dogs. Existing separation-related behaviours were reduced in lockdown as dogs were left alone less often and for shorter periods [7]. However, there are fears that these behaviours could resurface and become worse when more “normal” routines resume [8]. There could also be negative aspects of pet ownership during and after this period, as more time spent at home can increase the chances of aggression, especially with children [9], due to fewer opportunities for dogs to rest or isolate themselves. Parente et al. (2021) found that bites from familiar dogs to children, which required hospital admission, increased by 69% during 2020, and 72% of these were severe enough to require sutures. This significant increase is a potential concern for current and future dog welfare and also the safety of the community. During lockdown, some dogs were also walked less frequently, commonly on lead and closer to home, which can have negative welfare implications, causing stress and frustration, especially for dogs with high energy levels [10]. Dogs Trust found that reduced socialisation opportunities with people and other dogs throughout lockdown were the most common concerns [7]. There were fears that ‘isolation puppies’ would affect future dog walking routines due to a lack of socialisation. Increased stress and frustration can lead to the onset of undesirable behaviours [11]. This may affect how owners perceive their relationship with their dog, therefore potentially affecting the dog’s welfare, with 48% of dogs presented at the vet doing so due to fears and phobias [12]. Many dog owners have reported negative effects of lockdown on their pets, for example, concerns over changes in temperament and independence, becoming more ‘needy’ and frustrated [13]. Gray [14] found that 54% of study participants reported an increase in their dog hiding or moving away from people compared to pre-lockdown. Throughout COVID-19, many dogs’ welfare has been at risk as they have been threatened by novel everyday events and are therefore more susceptible to stress and disease [15]. These dogs may cause more stress and frustration to their owners; as a consequence, they are at higher risk of being abandoned, rehomed or euthanised [16]. Hoffman, Thibault and Hong [17] found that 8% of all owners had considered relinquishing their dog either during or after lockdown and 9% of dogs acquired through the pandemic were no longer with their owners, either being sold, rehomed or given to friends/family.

The present study was designed to explore the perceived changes in the social behaviours of adult dogs resulting from the implementation and subsequent withdrawal of COVID-19 regulations as well as perceived causes. This was undertaken in order to identify potential short- and long-term impacts of this event on pet dogs. Specifically, the study focused on interactions with people and other dogs. Research around this topic has been mainly focused on the social behaviour of puppies, rather than adult dogs that have experienced a novel change in routine. Studies on adult dogs have focused on the management of walking practices and separation anxieties that may be associated with lockdown. There is a clear gap in research identifying how adult dogs have adapted to lockdown, and whether there are any changes in their social behaviour during and/or after this period. By using qualitative research techniques, an in-depth insight can be gained into the perceptions and opinions of dog owners throughout the pandemic.

## 2. Methodology

### 2.1. The Qualitative Research Approach

Data were collected through qualitative methods using semi-structured interviews consisting of open-ended questions. The use of open-ended questions allowed for the exploration of topics in-depth, understanding practices and identifying the cause of any potential relationships [18]. A semi-structured interview format enabled the researcher to improvise any follow-up questions they may have had based on participant responses, and it also allowed for verbal expressions of different opinions [19].

### 2.2. Study Design

Due to COVID-19 restrictions, participants were contacted via Facebook from one of the authors’ friends list and community dog groups and invited to participate in semi-structured interviews via Microsoft Teams. Potential participants were provided with a brief explanation of the project and informed that interviews would be approximately 20 min. A consent form was also provided to participants which was co-signed and dated for the researcher’s records. Individual virtual face-to-face interviews were organised upon the response from each owner. Fifteen participants were chosen to satisfy data saturation levels [20,21]. Participants were chosen due to the ownership of a dog (aged between 3 and 6 years) before, during and after the COVID-19 lockdown. All participants lived in England throughout the COVID-19 lockdowns; therefore, restrictions would have been similar for all respondents. Data were collected between December 2021 and January 2022. Only dogs between 3 and 6 years of age were included in this study to avoid any changes to social behaviour due to adolescence or senescence. The criterion sampling technique was the most suited as it ensured that respondents had experienced the conditions which were relevant to the research [22]. Participants identified as both male (40%) and female (60%).

### 2.3. Interview Structure

Interview questions (Table 1) were formulated to investigate each owner’s personal experiences of social behaviour in an adult dog during and after the COVID-19 lockdown. One semi-structured interview was devised for all responders without a defined question order. This encouraged flow in each conversation and allowed interviewees to discuss topics in depth. Fifteen interviews were conducted by the researcher to help prevent any unnecessary bias or variation within the interview technique. At the start of each interview, a broad question based on the background and acquisition of the dog was asked. This was used to help interviewees to feel more relaxed and able to settle into the structure of the interview [23]. Once this initial question was answered, four structured questions were asked. These focused on different stages of the COVID-19 pandemic and any social behaviours observed. All interviews were completed and varied from 15 to 40 min, with interviewees keen to express their personal experiences. With participant consent, each interview was recorded via Microsoft Teams and uploaded to a secure laptop for transcription by the authors.

### 2.4. Thematic Analysis

Interviews were transcribed and analysed using NVivo 12 (QSR International; Melbourne, Australia). Thematic coding involved identifying passages of text that were linked by a common theme, allowing categories to be established [24]. This was used to establish key themes (nodes) within the interview. Categorising the transcripts into nodes relied on the continual reading of each typed interview and frequent editing. This ensured that theme development was checked adequately, and intra-coder reliability could be established [25]. Once the thematic analysis was completed, the nodes were analysed again, with the most relevant quotes being chosen for use within this study. A consensus approach was important for the implementation of evidence-based practice [24]. This involved a group process, which included H.B. coding and analysing the data and M.J.F. providing supervision and feedback, followed by editing from H.B. and M.J.F.

### 2.5. Ethical Considerations

Before any data were collected, an experimental protocol was developed, and ethics approval was granted by Harper Adams University (project number 0552-202110-UNDER).

## 3. Results

Twenty-two owners expressed interest in participation. Six subsequently declined to be interviewed and one expressed interest only after the study had closed due to data saturation being reached. The study sample included a range of dog breeds who varied in age and where they were acquired (Table 2). Fifteen interviews were conducted with owners who discussed the effects of lockdown on 18 dogs. To illustrate themes and sub-themes, the interview responses are presented as quotes below, with the respondent number provided in brackets, corresponding with the information in Table 3.

Following thematic analysis of the interview data, four key themes were identified: changes in environment, changes in dog–dog behaviour, changes in dog–human behaviour and changes to non-social behaviour. Sub-themes and perceived outcomes were also identified; these are shown in Table 3.

Further illustrative quotes are used to expand on the theme. Other relevant quotes can be found in the Appendix A.

### 3.1. Changes in Environment

Variations in environment during lockdown were reported by several respondents. These included external and at home changes. Five respondents cited that their walking routes were busier than usual; however, three respondents noted that walks were quieter. Three owners also perceived that their dogs had an increased duration and frequency of walks due to working from home and having time to spare.


*“Walks were quieter than before lockdown. You didn’t see as many people where we used to.”*
—*(R1)*


*“I would go on walks in the countryside, and I could bump into groups of people walking. And that’s never usually what happens around here, normally it’s maybe the occasional dog walker, and a bike.”*
—*(R15)*

The shift between conventional and remote work affected ten households, with either one or both owners having different routines. This resulted in increased owner presence in eight households.


*“Before COVID I worked from home probably two days a week. And it changed week on week so there was never a formal [amount of time] that she was always on her own and then when the pandemic hit, I worked from home all the time, and so did my wife. So we were both at home all the time.”*
—*(R5)*

The lockdown did not affect all routines in the same way. A small number of owners explained that due to their usual changing schedules, their dogs did not experience uncommon routines.


*“I still have days at home now, but I’ve always done those odd days with my job. I start early and I could [be] home in the middle of the day and then back out. So they’re never on their own long.”*
—*(R1)*


*“I work from home so she’s barely on her own. I think I might have always worked from home since we’ve had her or if not, it wasn’t long after getting her, I’ve always worked from home, so she’s pretty much used to it.”*
—*(R7)*

### 3.2. Changes during Lockdown

Positive changes were identified by several respondents. These included changes to dog–human and dog–dog social behaviour. Two cited positive changes such as increased dog social skills and three respondents noted their dogs had been calmer due to owners being at home for the majority of the time.


*“She seemed to appreciate time with other dogs more if that is something that you can ever actually articulate.”*
—*(R5)*


*“She coped really well [with the new puppy]. It’s the usual thing with the overexcited puppy jumping on her, but they established an order and there was no aggression. She welcomed [the puppy] in the house.”*
—*(R6)*


*“When lockdown first happened, I would say her reaction towards us was better as in she was more involved and wanting to be around us. So if I was upstairs working, she would want to come up and see me or if my wife was downstairs doing something, she would go and see her.”*
—*(R5)*

Negative changes to dog–human and dog–dog social behaviour were also identified. Three respondents cited attention-seeking behaviours and three noticed increased reactivity to passing dogs and walkers.


*“When I went from working quietly to being on a Teams meeting, he comes in and always puts his nose on the laptop because he can hear people and he can hear me talking obviously. That’s when he’s a bit more attention-seeking and then he’ll do something naughty to get my attention.”*
—*(R2)*


*“Her routine changed because just before lockdown, my older dog died. Then I couldn’t leave her because she got really stressed at being left on her own. So she became a bit clingy.”*
—*(R8)*


*“He’s gone from barking when people are on the drive, to doing it when people are walking by our house. It could just be someone taking the kids to school and he’s barking at them. He hates the postman now and I don’t know if that’s just because he was one of the only people coming to our house. His hackles go up when the postman is there, which is weird because he’s the nicest friendliest dog ever.”*
—*(R2)*

### 3.3. Changes after Lockdown

Positive changes to dog–dog and dog–human social behaviour were identified by several respondents. One owner cited less excitement toward other dogs, and three owners noted their dogs were calmer when people came to the house, whether they were guests, people making deliveries or people walking past the window. Some owners believed these positive changes were solely due to age (R9).


*“They’re less bothered [by other dogs], but that might be just an age thing. They’ll notice and maybe bark but then carry on with us as opposed to running over.”*
—*(R11)*


*“She’s perhaps a little better behaved when we’re opening the door. If you tell her to stay she quietens down and stays and doesn’t always dart for the door to get out. Possibly because you get a lot more deliveries, food delivered and Amazon.”*
—*(R7)*


*“Guests coming to the house was slightly better. I think I personally attribute that to her age. The lockdown was 15 months or thereabouts. So she almost doubled in age during that period. So she was at a calmer age by the time the lockdown finished as well.”*
—*(R9)*

However, many respondents identified negative changes to dog–dog social behaviour. Two owners’ dogs experienced dog attacks which may have increased fear, and three owners cited an increase in reactive behaviours, with one noting aggressive behaviours, e.g., snapping.


*“One of the dogs [we passed on our walk] for whatever reason just took a dislike to [my dog]. My son had to literally prise this other dog’s jaws off [my dog]. This woman was ever so apologetic. It was her dog that started it, but my dog retaliated.”*
—*(R4)*


*“Since lockdown he has snapped at other dogs. I can normally judge and we’ll either cross the road or they’ll have crossed the road when they’ve seen he’s coming. So you could avoid it. But even then, if he’s the other side of the street [to other dogs], I will bring him close and get the lead across me and make sure he can’t get in front of me. Because I use a retractable lead and they’re a bit harder to control when you need to wind him in. I guess it’s a defence mechanism. I tend to think it’s kind of learned behaviour from once you’ve got one bad experience with a certain type, he’s waiting for that problem to happen again. And he wants to get in first for protection really.”*
—*(R3)*

Others noted negative changes in dog–human social behaviour. Eight owners noted that their dogs were excited around guests coming into their home and seeing people on walks, three cited increased reactivity and three owners noted increased fear and caution.


*“When people started coming back to the house, they were really gone. They’ve gone over excited and jumping at people. They’re not used to people coming.”*
—*(R1)*


*“Whether she would still be as giddy if we hadn’t had lockdown, I don’t know. But we noticed that once we were allowed to see people again, she was so happy. She must have felt the effects of it as well.”*
—*(R4)*

One owner found that they were not able to socialise their newly rescued dog with babies and young children throughout lockdown. Consequently, their dog had a negative reaction towards their newborn child, which resulted in the dog being relinquished.


*“On the first day we came home from the hospital we tried to keep [our dog] at home for as long as we could, but she was showing signs of basically wanting to nip and play [with the baby]. But not in like a friendly nip and play it was more like a dog seeing a squirrel up a tree, that sort of mentality. So, she then went into kennels for about a week and then had an assessment with Dogs Trust.”*
—*(R5)*

One owner noted that separation-related behaviours had improved, and the dog was able to settle in the owner’s absence; however, another owner found that these behaviours had worsened due to undivided attention from them.


*“If we do tend to leave the house now, there’s no barking. Because I think she’s learned we do come back. Now she stands at the door, and you can see her until you drive off. But then she’ll go find somewhere quiet to have a lay down so that’s key.”*
—*(R14)*


*“She was downstairs on the bottom floor. She was down there for a while actually, just crying. It stopped after a while but that’s a bit different from where it was before. My wife’s never left the house through lockdown and now when she does, [my dog is] a little bit more anxious, I’ve noticed that.”*
—*(R6)*

### 3.4. Changes to Non-Social Behaviour

Negative changes to non-social behaviour were also identified. Four respondents cited training regression, with two owners requiring behavioural intervention post-lockdown.


*“I did try to do training with him, sometimes I wouldn’t see anybody and then when I wasn’t expecting to do training with him, I’d see people. So I could never be prepared for what was going to happen in one day.”*
—*(R15)*


*“We went to a dog behaviourist when he was around six months old, and we got a one-to-one behaviour modification course. He regressed in everything during lockdown, but it’s pretty easy to get it back now we know what to do.”*
—*(R2)*


*“I’ve had to have a behaviourist out because of the anxiety issues.”*
—*(R1)*

Other respondents cited fear-related changes in their dog’s non-social behaviours. One cited an increased fear of motorbikes and five cited an increased fear of car journeys. This was possibly caused by a reduction in travel due to lockdown restrictions.


*“He is terrible with motorbikes. They were trying to go on the trails where I would expect them every now and again, but it was to the point where I had to avoid that trail because they’re coming here a lot.”*
—*(R15)*


*“She’s not very good in the car. She’s sick in the car. It has probably got worse because obviously she’s not in the car as much. Previous years she’s not sick every time. But this time, she’s been sick every time.”*
—*(R7)*

### 3.5. Future Concerns

Five owners admitted they had concerns relating to the impact going back to their normal routine and going on holidays abroad would have on their dogs. One owner felt that if lockdown did not happen, they would have been able to socialise their anxious dog more, and one owner felt that behavioural changes seen through lockdown were coincidental and only occurred due to their dog’s age.


*“I’d worry about going back to work full time. And if my husband didn’t do shifts, about leaving them on their own for a full day. I think they would maybe struggle with that now because I can’t think of the last time that happened.”*
—*(R11)*


*“We could have spent a lot more time with other people around. She was starting to get better and become accustomed to different people that are coming in and out of the house. We might have even looked to get a second dog and getting that family unit set up beforehand. I think it certainly would have improved her chances.”*
—*(R5)*


*“I think it was entirely coincidental. I think even if there had been no lockdown and I continued putting them in kennels every now and then. I think really, we’re just seeing no difference.”*
—*(R9)*

## 4. Discussion

Given the impact of the COVID-19 pandemic on people’s lives, it was expected that this would also be reflected in pet dogs. With lockdown restricting daily movement, activities and social interactions between people, it was hypothesised that lockdown may result in more behavioural changes in adult dogs as reported by owners. To the authors’ knowledge, this is the first qualitative investigation into changes in the social behaviour of adult pet dogs throughout the COVID-19 pandemic. The qualitative approach of this study gives an understanding of the behavioural changes that owners have witnessed throughout this period and why they believe these changes may have occurred. In-depth discussions were used because “a qualitative approach is especially appropriate when little is known about a topic” [26]. A purposive sampling method was used, allowing for the aims of the study to be explored, to reach sufficient depth of statements for significant analysis and to give the interviewee the ability to introduce and discuss other topics relevant to the study aims. This would be difficult to achieve with quantitative methods [27].

Many owners appeared to consider the emotions of their pet dogs when reviewing the impact of the pandemic on their companions. This showed owners felt connected with their pets through emotions and feelings. Range and Virányi [28] suggest that dogs are responsive to human emotional expressions. It is possible that additional human stressors present during the COVID-19 pandemic [29] caused concomitant changes in companion dog behaviour. This may be shown by owners stating their dogs became more “clingy” throughout this uncertain period. Some participants noted that before COVID-19, their lives limited the time they spent with their dogs. Due to lockdown restrictions, they were able to spend more quality time with their dog and, as a whole, participants enjoyed the additional time spent with their companion. However, these interactions may have negative long-term effects as increased dog–owner interactions could result in increased separation issues, as seen in “pandemic puppies” [3].

Socialisation in dogs is most commonly researched at a young age due to the effects of sensitive periods on the development of behavioural problems. This study is different in that social behaviour has been studied in adult dogs experiencing a drastic change in environment. Titulaer et al. [30] found that dogs that had been kennel housed for more than six months were more likely to show aggression towards other dogs and were less likely to interact with strangers. Some respondents reported similar findings within the current study, with nine owners observing an increase in aggression and fear-related behaviours. An increase in the duration of walks in this study is comparable to the findings of Owczarczak-Garstecka et al. [11]. However, the current study suggests that many owners continued to walk their dogs more than once a day, even when government restrictions were clarified. This was clearly due to the negative impact that the restriction would have on their pets. Evidence suggests that increased walks are associated with fewer dog behaviour-related problems [31]. Similarly, Shoesmith et al. [15] observed that many dogs became more attached to owners, with the words “clingy” and “needy” appearing commonly. This was described as having a negative impact on the animal’s temperament; however, many respondents in this study felt that it was a positive experience for their pet as they had the attention they desired.

There is a very real risk to human and animal safety as displayed in the results of this research. Many owners observed fear-related behaviours, which could escalate to aggression if not managed correctly. The UK is already seeing an increase in paediatric dog-bite emergency department attendances [32]. Owner education is needed urgently to raise awareness of the dog bite risk to humans and animals—especially to children. Advice from owner education programmes includes how to approach and interact with dogs, how to recognise body language and how to keep safe if approached by an unknown dog [33]. Safe interactions should be promoted within and outside of the household. It can be seen in this study that the change in environment and restrictions during the COVID-19 pandemic have induced some aggressive behaviours or exacerbated existing behavioural issues. These variations may have occurred without the influence of the pandemic, but possibly at a slower rate. Relinquishment of dogs is high due to the pandemic, suggesting that there is a risk to the dog’s behaviour, health and welfare in some cases. One interviewee had to surrender their dog due to aggression around their newborn child and was certain that there would have been more opportunities for a different outcome if COVID-19 restrictions were not in place. It is predicted that 40,000 dogs will be in danger of relinquishment as a direct result of the pandemic, as owners struggle to cope with problematic behaviours [34].

There is a growing need for behavioural support as dogs struggle to cope with social changes due to the pandemic. Two owners in this study enlisted the help of a dog behaviourist to help them to manage their dog’s new fear-related behaviours, with other owners commenting that they may need similar support. Behavioural therapy conducted by a qualified professional can support owners in dealing with new problem behaviours. The use of positive reinforcement is associated with better obedience and learning ability. However, the use of aversive-based techniques leads to more aggression and stress-related behaviours [35]. There should be an increased provision in public-facing resources for this cohort as well as owners with “pandemic puppies” to manage and alleviate the effects of inadequate socialisation throughout this period. Many owners feel there is a stigma attached to dogs that show aggressive behaviours and feel “embarrassed” by this. They will therefore feel reluctant to seek help from professionals; because of this, they risk the escalation of behaviours. Behavioural intervention should focus on the physical and behavioural environment [36], which involves alteration of the environment alongside behaviour modification. Dietz et al. (2018) [37] found that expert advice on dog behaviour appears to be effective in reducing the prevalence of behavioural disorders and should be given to owners proactively.

Future implications of the COVID-19 pandemic are unpredictable and may result in a change in employment or working location for many owners over the coming weeks, months or years. Owners should feel confident that their dogs can cope with these changes if they occur again. Results from this study may also be applied to owners that have experienced a period out of work with altered routines, as separation-related effects would be similar in these cases. Brand et al. [3] similarly found that there were negative implications of separation shown; however, with this research focusing on adult dogs, it was positive implications were also identified, with some dogs able to cope better when left alone after lockdown. This study may also be comparable when owners move property, whether this is from a quiet to a busier area or vice versa. Some dogs in the study appreciated more contact with people and dogs after a quieter period, whereas others struggled when their regular walking routes were busier. The long-term implications of lockdown on dog social behaviour are still unknown. Further research is needed to investigate long-term behaviour changes to further assist owners in caring for their pet dogs and to prepare dog behaviour professionals for future problems.

This study is important because it examines dog behaviour from the owner’s point of view. The semi-structured nature of the interview provided the interviewee with an opportunity to discuss other topics relevant to the study, ensuring relevant information was not missed due to structured questioning. Although these findings may not be representative of the entire dog-owning population, the number of participants interviewed did allow for data saturation. Therefore, there is confidence that major social behaviours were identified through emerging key themes. Further studies may consider identifying whether there are any differences in the wider population or any geographical variations. Whilst these themes may not represent all opinions of dog owners, both positive and negative owner opinions were sought.

Nonetheless, limitations to the research exist. Asking owners their thoughts and opinions may cause a self-reporting bias. This occurs when participants provide self-assessed measures of a specific topic [38]. Social desirability bias is the underreporting of attitudes due to being regarded as socially undesirable [39]. This occurs frequently when interviewees are questioned regarding personal or sensitive topics. The study aims were not to confirm claims made by respondents; however, signing the consent form confirmed confidentiality and anonymity, which decreases the likelihood of this bias [40]. Recall bias occurs when interviewees cannot remember past experiences accurately or recollect detail [41]. As this study researched opinions over 18 months, detail may have been impacted by respondents’ information recollection from the start of lockdown. Quantitative inter-coder reliability (ICR) is often identified as a vital component to qualitative research; although this is not always the case [42,43], this is a limitation and potential source of bias in the current project. Herein, informal processes were used between the co-authors to reach consensus around the themes represented. However, all initial coding was conducted by a single researcher who is embedded within the field of dog–owner relationships and, beyond informal agreement of themes based on quotes/outputs, it was felt that ICR in a small research team may, in fact, detract from this personal grounding and interpretation. In the discussion we have triangulated the findings with the wider literature, identifying common themes in the work of others which, in part, lends credence to the themes arising within this work. These results could be considered exploratory, requiring further validation in future studies.

## 5. Conclusions

Understanding the impact of COVID-19 on dog behaviour is crucial to enhancing dog and owner welfare throughout these periods and similar circumstances. Few, if any, previous studies have attempted to recognise the effects of lockdown on owner perception of social behaviour in adult dogs and why changes may have occurred. To achieve a more comprehensive understanding of owner experience, a larger sample size over a longer period could be used. Additionally, by including an equal variation of dog breeds or ages, there would be a possibility of identifying any breed/age specific behaviour changes during this period.

This project aimed to determine whether there were any changes to dog social behaviour during and after the COVID-19 lockdown and why these changes occurred. The results of this study illustrate that many households have faced different experiences. Owners regarded lockdown as a positive experience for their pet dogs; however, many dogs faced difficulties when restrictions were lifted. Emphasis was largely placed on the negative impact of lockdown, with most dogs becoming either overexcited with people or showing fear-related behaviours towards people or dogs. The majority of owners felt that their dog’s behavioural changes did occur due to a change of routine and the introduction of lockdown restrictions. A small number of owners felt that the age of the animal contributed to the positive changes they had seen.

To conclude, there is a variation of different views and opinions regarding dog behaviour changes throughout the pandemic. More awareness is needed of the safety risks of improper socialisation. Education is key in preventing these unwanted behaviours and behavioural intervention should be sought. Findings can be used to inform owners and dog welfare strategies for future pandemic situations or in other disasters and emergencies likely to impact daily routines. Further research into the long-term implications of the pandemic on dog social behaviour is required. In doing so, the socialisation of adult dogs can be understood more thoroughly, and behavioural support can be tailored suitably.

## Figures and Tables

**Table 1 animals-12-01682-t001:** Interview schedule.

Semi-Structured Interview Questions
Please explain your work situation through COVID-19. Have you been working from home, in the office as usual, or working more as a key worker?
What was your dog’s ‘usual’ socialisation behaviour with dogs and people before COVID-19?
During the COVID-19 lockdown, did you see any changes in your dog’s socialization behaviour towards other dogs or people? Can you describe the behaviour that was shown? Why do you think these changes have occurred?
Following the COVID-19 lockdown, did you see any changes in your dog’s socialization behaviour towards other dogs or people? Can you describe the behaviour that was shown? Why do you think these changes have occurred?

**Table 2 animals-12-01682-t002:** Interviewee gender and information about their dog(s).

Respondent Number	Respondent Gender	Age of Dog	Breed	Sex	Age Acquired	Where Acquired
1	F	4 years old	Springer Spaniel	Female	9 weeks	Given
		6 years old	Springer Spaniel	Female	8 weeks	Breeder
2	F	3 years old	German Shorthaired Pointer	Male	8 weeks	Breeder
3	F	4 years old	Maltese Terrier	Male	6 months	Advertisement
4	F	4 years old	Springer Spaniel × Poodle	Female	8 weeks	Advertisement
5	M	4 years old	Bali Heritage Dog	Female	6 months	Rescued from Bali
6	M	10 years old	Labrador × Poodle *	Female	8 weeks	Advertisement
		4 years old	Miniature Dachshund	Female	8 weeks	Advertisement
		1 year old	Miniature Dachshund *	Female	8 weeks	Advertisement
7	M	4 years old	Bichon Frise × Shih Tzu	Female	10 weeks	Advertisement
8	F	4 years old	Jack Russell × Pomeranian	Female	8 weeks	Advertisement
9	M	3 years old	Unknown breed	Female	12 months	Rescued from Iran
		3 years old	Border Collie	Male	13 months	Rescued from Ireland
10	M	3 years old	Miniature Schnauzer	Male	12 weeks	Advertisement
11	F	5 years old	Lhasa Apso	Male	8 weeks	Breeder
		5 years old	Lhasa Apso	Male	14 weeks	Rescued
12	F	3 years old	Staffordshire Bull Terrier	Male	8 weeks	Breeder
13	F	3 years old	Cocker Spaniel × Poodle	Male	8 weeks	Advertisement
14	M	6 years old	Maltese Terrier	Female	8 weeks	Breeder
15	F	6 years old	Parsons Jack Russell × Border Collie	Male	12 months	Rescued

* Discounted from the study due to age.

**Table 3 animals-12-01682-t003:** Primary and sub-themes, perceived outcomes and quotes concerning changes in adult dog behaviour as a result of COVID-19 lockdowns in England identified through semi-structured interviews with participants (*n* = 15).

Major Theme	Sub-Themes	Perceived Outcomes and Participants Reporting (*n*)	Quotes
Changes in environment	External changes	Increased walking frequency (4)	*“Even in the first part of the lockdown, he got walks every day. Long walks as well, rather than, I’ve got half an hour in between meetings, I’ll take him out for a quick walk, not a long walk. So he was loving it all the way through lockdown because he had someone at home looking after him.”—(R10)*
		Busier walking routes (5)	*“There’s one particular location I will go to where suddenly in March 2020 it’s almost like the whole of [city] suddenly discovered this place and it went from being somewhere which was really not very busy to somewhere which was always really busy.”—(R9)*
		Quieter walking routes (3)	*“In the initial part of lockdown, it was much quieter. And I think that was mainly because people were unsure, didn’t want to go out and wasn’t sure what they were allowed to do.”—(R10)*
	At home changes	Remote working (10)	*“Yeah, I used to work in the office probably three days a week. My wife does stay at home, but she would need to nip out every now and again. So it was at that point that she’d be on her own. Now, I’m very rarely in the office. You know, kind of been in three times I think in the last 18 months.”—(R14)*
		Increased owner presence (8)	*“I used to work in the office four days a week and one day at home. Pre-COVID my wife used to work only half days. So he was only at home on his own for about three and a half hours on those days that I wasn’t here. So he was loving it all the way through lockdown because he had someone at home looking after him”—(R10)*
Changes in dog-dog behaviour	Increased reactivity	Dog attack (2)	*“[My dog] had an incident where we were on the field and a dog ran up to her and she panicked and actually ran home from across this main road, and we were screaming.”—(R1)*
		Increased fear (5)	*“His reactivity got worse, because he had quite a few bad encounters with off lead dogs, so he tended to get very worried by other dogs especially when they start getting giddy and excited.”—(R15)*
		Increased aggression (2)	*“Before lockdown, we’d be able to walk anywhere, and he would always be so happy to see dogs. Now when we take him around the block, there’s a couple that he will bark at in a nasty way. And he never used to be like that.”—(R13)*
	Decreased reactivity	Spent more time in the company of dogs (2)	*“She seemed to want to play with more dogs than she would have done previously.”—(R5)*
		Less excitement toward dogs (1)	*“I know pre-lockdown they’d be overexcited about seeing another dog and definitely have to go over and say hello. And if the dog was on a lead, we’d be constantly putting them back on the lead, so they don’t run over. Whereas now they do tend to stick with us more.”—(R11)*
Changes in dog–human behaviour	Fear-related changes	Increased anxiety (4)	*“If she doesn’t know someone, she stands off until I’ve spoken to them or occasionally, she’s barked. It was an ‘I’m not too sure’ bark. And that’s only happened since lockdown.”—(R8)*
		Increased reactivity (3)	*“I think barking has got worse to be fair, at everyone.”—(R2)*
		Decreased reactivity (3)	*“Guests coming to the house was slightly better.”—(R9)*
		Relinquishment (1)	*“[My dog] wasn’t constantly barking because it’s a human [baby], it was more ‘Can I play with this’. So that’s the reason why we’ve ended up having to rehome her.”—(R5)*
	Separation-related changes	Increased separation distress (1)	*“My wife’s never left the house through lockdown and now when she does, [my dog is] a little bit more anxious, I’ve noticed that.”—(R6)*
		Decreased separation distress (1)	*“If we do tend to leave the house now, there’s no barking.”—(R14)*
	Other changes	Increased excitement (4)	*“Now [when visitors arrive] it’s completely over the top because she jumps into people’s arms.”—(R4)*
		Increased attention-seeking (4)	*“He did want attention more* *.”—(R2)*
		Increased calmness (3)	*“He has generally been more chilled out.”—(R2)*
		Companionship (6)	*“He just had companionship all the time for 24 h a day, so I think he was pretty in his element with lots of fuss and attention.”—(R3)*
Changes in non-social behaviour	Training and behavioural intervention	Training regression (4)	*“He regressed in everything during lockdown, but It’s pretty easy to get it back now we know what to do.”—(R2)*
		Involvement of canine behaviourist (2)	*“* *We actually paid for a dog behaviour therapist to come around and give us some tips about how to get people in the house.”—(R5)*
	Fear-related changes	Fear of motorbikes (1)	*“He is terrible with motorbikes. They were trying to go on the trails where I would expect them every now and again, but it was to the point where I had to avoid that trail because they’re coming here a lot.”—(R15)*
		Fear of car journeys (5)	*“She was quite nervous in the car. And would sit a bit shaky, squealy because previously before lockdown she really only went in the car when we were going on a journey because I used to walk her locally.”—(R8)*
Future concerns	Separation-related problems	Holidays (1)	*“It does worry me a bit because next year we’re thinking we want to go on a foreign holiday. But then I’m worried because obviously [my dog is] used to us being [with her].”—(R4)*
		Return to conventional work (5)	*“You don’t realise when you’re at home with them all the time then suddenly you’re not, what difference it makes.”—(R1)*

## Data Availability

The anonymized data presented in this study are available on request from the corresponding author.

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
