# Peer review of "Changes to Adult Dog Social Behaviour during and after COVID-19 Lockdowns in England: A Qualitative Analysis of Owner Perception"

_animals, 2022, doi:10.3390/ani12131682_

Round 1

Reviewer 1 Report

This ms reports results from a qualitative study about dog owners' perceptions of their dogs' behaviour during and after COVID lockdowns. This is  a useful contribution to the literature because anecdotal evidence of dog behaviour problems post-lockdowns are becoming more common. It's important to systematically investigate this. Before I recommend it for publication, I have some suggestions.

Major comments

1. The data analysis section needs more detail. Most importantly, how can inter-rater reliability be confirmed if only one person did all of the analyses? This is a potential dealbreaker so it needs to be explained clearly. Other specific comments below.

2. The results are presented in an awkward way. It is strange to have a short paragraph briefly describing each result and then a list of several quotes around that topic. I recommend that the authors do one or more of the following instead: create a column within Table 3 that has an illustrative quote for each perceived outcome; pick one or two illustrative quotes for each of the topics, rather than several; describe the quotes better with more integration of the main text and the quotes. Any combination of these will improve flow and make the piece more concise. The formatting as it stands is also weird because there are lists of quotes, and then an explanation with no line breaks, which makes it hard to see.

Specific comments

Title - shouldn't the title say covid-19 lockdowns (plural) instead of lockdown (singular)? I believe there have been more than one now in the UK

L13 - add 'years' after 'between 3-6'

L17- in which context(s)?

L21 - add 'during COVID-19' at the end of the sentence.

L23 - why are two aims listed? It doesn't look right. Suggest combining them into one aim.

L24 - was Scotland included as well? Or just England? Where specifically in the UK? I ask because I believe the lockdown restrictions varied from country to country within the UK. Please clarify either way.

L28 - what period? This is not clear in the abstract or the main text. The data collection period was not described and it should be available in both places

L28 - behaviours increased when? during lockdown or after? clarify. Many people will (unfortunately) only read the abstract so the key points need to be made here.

L46 - please change 'disabled' to person-first language, 'people with a disability'

L53 I believe the right word here is 'consequently'. consequently = happening because of. subsequently = happening afterwards

L65 - please provide more info about this statistic because it is very alarming, and frankly seems unrealistic. Is this referring to reported bites? Bite victims that ended up in hospital? I find it hard to believe that nearly 3/4 of ALL dog bites end in sutures.

L75 - dogs referred to the vet by whom? Do the authors mean dogs presenting at the vet?

L102 - suggest moving the table just below this para, or removing this reference (the latter is probably simpler and would flow better).

L115 - data saturation levels - were these determined a priori or post-hoc? If a priori, on what basis?

L115 - when were data collected?

L116 - which countries within the UK were participants from? Were restrictions all the same for all participants?

L134 - transcribed by whom? A company? the authors? an RA?

L142-144 - this should go at the top of the Results section.

Table 2 - suggest changing 'advert' to 'advertisement' or 'ad'

L204 - 'never a formal between now and then' doesn't make sense. I understand that it is a quote but it could be altered slightly, or a word inserted in [brackets], to improve readability

L221 - as above re: readability of the quote '...I could home in the middle...'

L236-237 - more context is needed for this quote. who welcomed whom into the house?

L254-257 - the authors put this quote in the negative category, but the way the speaker describes it doesn't sound like they consider it to be negative.

L274 - heckles should be hackles

L280-284 and L291-293 - the authors should mention that these participants believed the changes were due to age and nothing else. That is not mentioned at all in L277-280.

L298 - took a dislike to whom? Who is the other dog? More context needed.

L302-305 - the participant clearly mentions that this is coincidental and not to do with the COVID restrictions.

L328 - consequently would be better than subsequently here, too

L336-339 does not appear to be covid-related.

L347 - 350 - this needs more context. It seems very serious. What was the outcome? What led to this?

L490-491 - awkward phrasing, it sounds like the authors mean that the owners will feel reluctant to risk escalation of behaviours, but what I think they mean is that not seeking help will risk escalation.

L527 - please cite the claim that identity protection will reduce the risk of biased responses.

Reviewer 2 Report

Please provide more information on the process used for obtaining respondents for the study. What social media did you use?  Did you  advertise to groups associated with dogs?

in regards to dog training and behavior specialist s which method of training used?  Methods that utilize negative or pain based may increase aggression and anxiety.  Were any dogs treated with meds prior to the pandemic? 
Did any of the participants relinquish their pets?

How might increased walks during the pandemic impact behavior?

How did you determine saturation?  Did you examine gender differences in reporting behavior changes in participants pet

Round 2

Reviewer 1 Report

The authors have done a very good job of tightening up the paper and clarifying some confusing elements. I am happy to accept it for publication.